# Effect and Mechanisms of Antibacterial Peptide Fraction from Mucus of *C. aspersum* against *Escherichia coli* NBIMCC 8785

**DOI:** 10.3390/biomedicines10030672

**Published:** 2022-03-14

**Authors:** Yana Topalova, Mihaela Belouhova, Lyudmila Velkova, Aleksandar Dolashki, Nellie Zheleva, Elmira Daskalova, Dimitar Kaynarov, Wolfgang Voelter, Pavlina Dolashka

**Affiliations:** 1Faculty of Biology, Sofia University, 8 Dragan Tzankov Blvd., 1164 Sofia, Bulgaria; mihaela.kirilova@uni-sofia.bg (M.B.); baba_emi@abv.bg (E.D.); 2Institute of Organic Chemistry with Centre of Phytochemistry, Bulgarian Academy of Sciences, Acad. G. Bonchev Str., Bl. 9, 1113 Sofia, Bulgaria; lyudmila_velkova@abv.bg (L.V.); adolashki@yahoo.com (A.D.); mitkokaynarov@abv.bg (D.K.); 3Faculty of Physics, Sofia University, 5 James Bourchier Blvd., 1164 Sofia, Bulgaria; zhelevan@phys.uni-sofia.bg; 4Institute of Biochemistry, University of Tübingen, Hoppe-Seyler-Straße 4, D-72076 Tübingen, Germany; wolfgang.voelter@uni-tuebingen.de

**Keywords:** mucus of *Cornu aspersum*, peptide fraction MW < 10 kDa, *Escherichia coli* NBIMCC 8785, SEM, fluorescence and digital assays, antibacterial effect

## Abstract

Peptides isolated from the mucus of *Cornu aspersum* could be prototypes for antibiotics against pathogenic bacteria. Information regarding the mechanisms, effective concentration, and methods of application is an important tool for therapeutic, financial, and ecological regulation and a holistic approach to medical treatment. A peptide fraction with MW < 10 kDa was analyzed by MALDI-TOF-TOF using Autoflex™ III. The strain *Escherichia coli* NBIMCC 8785 (18 h and 48 h culture) was used. The changes in bacterial structure and metabolic activity were investigated by SEM, fluorescent, and digital image analysis. This peptide fraction had high inhibitory effects in surface and deep inoculations of *E. coli* of 1990.00 and 136.13 mm^2^/mgPr/µMol, respectively, in the samples. Thus, it would be effective in the treatment of infections involving bacterial biofilms and homogenous cells. Various deformations of the bacteria and inhibition of its metabolism were discovered and illustrated. The data on the mechanisms of impact of the peptides permitted the formulation of an algorithm for the treatment of infections depending on the phase of their development. The decrease in the therapeutic concentrations will be more sparing to the environment and will lead to a decrease in the cost of the treatment.

## 1. Introduction

The discovery and testing of new therapeutic tools with a studied and clear effect on different bacterial infections is becoming increasingly topical. The requirements for using such kinds of therapeutic tools are also increasing [1,2,3,4,5,6].

The reason for this search in the first place is that the antibiotics that have been widely used previously led to the development of resistance by pathogenic microorganisms. The new tools should be such that their bio attacks on pathogens do not lead to resistance in the pathogenic microorganisms [7,8].

Two consequences related to the epidemiologic and therapeutic measures have appeared: (1) due to the wide use of antibiotics in different combinations and treatment sequences, the danger of the appearance of resistance to contemporary antibiotics and infectious pathogens that represent a serious medical risk for people and animals has been accelerated; and (2) the low biodegradability and wide distribution of antibiotics have led to their accumulation in civil wastewaters, which has led to another serious ecological risk [3,9,10].

The ecological consequences are immeasurable compared to the medical ones and from a long-term perspective. The antibiotics and their metabolic products are present in purified waters and are distributed in water basins, drinking water, plants, and animals, and through the trophic chains, are returned to humans. These pollutants in the wastewaters and in the environment combine and interact with other chemical factors. All these factors inhibit the activated sludge and microorganisms that purify the waters and lead to a lasting disturbance of the restorative processes of the resources (waters, soils, biomass, sediments, etc.) in the treatment technologies for the environment as a whole [3,4,9].

These combined interactions with human health and life and the health and future of the planet are lasting. In the future, they will impact strongly on the life of the planet [3]. The sources of the described dangers should be limited and discontinued, and intelligent solutions to the problems of the people, health, and the planet in the whole life cycle of therapeutic tools and of antibiotics should be found. The roads are three: (1) creation of new technologies for water treatment, and natural resources with included manageable detoxifying elements directed to the degradation of the toxic pollutants present and incoming in the environment; (2) replacement of the toxins in environment pollutants with their harmless alternatives; and (3) minimization of the tools used for medical and other aims that have harmful impacts on the environment from a long-term perspective. The combined consideration of the questions regarding therapeutic tools, effective concentrations, objective and personalized impacts, the life cycle, and impact on the environment and on the planet require a holistic approach and a deep understanding of the mechanisms of impact, distribution, and removal. This understanding is the basis of the concept of a less-toxic environment ensured by innovative therapeutic products and clean technologies for the sustainment of all resources [1,3,9,11,12].

The discovery of therapeutic products with an antimicrobial action and with a chemical structure close to natural compounds is a contemporary tendency and is in unison with the concept that “Nature knows best—follow it”.

The World Health Organization (WHO) declared that the increase in antimicrobial resistance to conventional antibiotics in recent decades is one of the three global threats to human health worldwide, and announced the start of the “post-antibiotics era” [13,14].

Invertebrates lack the adaptive immune system that is found in vertebrate species; therefore, they rely solely upon their innate immunity, in which AMPs play a crucial role in counteracting invading pathogens [15,16,17,18]. Molluscs are a large natural sources of AMPs, in part because of their great diversity (superior only to arthropods) and their ability to adapt to almost all habitat types [17,19,20,21,22,23].

AMPs hold considerable potential for reducing and overcoming antibiotic resistance because they not only have an antibacterial effect, but they also increase membrane permeability, and as a consequence, their presence enhances the effect of the antibiotics [24]. To adapt to this, bacteria need to develop major changes in the structure and properties of their membranes [11]. Due to their broad spectrum of action, natural AMPs can be a model for the discovery of new antimicrobial drugs that can address the problem of multidrug resistance of pathogenic microorganisms [25]. Currently, a number of bioactive compounds from the hemolymph and mucus of snails are being studied and developed as pharmaceuticals and/or raw materials for therapeutic purposes and applications [18,19,21,22,26,27]. For example, the garden snail (*Cornu aspersum*) and the giant African snail (*Achatina fulica*) are amongst the most studied land snails [18,27]. The water gastropods also offer a variety of AMPs. Some of the biologically active molecules identified were developed into pharmaceuticals such as Prialt^®^, Adcetris^®^, and many others [28].

Several peptides from the hemolymph of the garden snails *H. lucorum* and *H. aspersa* have been reported to show a broad spectrum of antimicrobial activity against *Staphylococcus aureus*, *Staphylococcus epidermidis*, *E. coli*, *Helicobacter pylori*, and *Propionibacterium acnes* [22,26]. The mucus content of snails varies depending on the function and secretory structure of the glands and species of the gastropods [29]. Zhong et al. found the AMP “mytimacin-AF” (9700 Da) in *Achatina fulica* mucus; it exhibited antimicrobial activity against *S. aureus*, *Bacillis* spp., *Klebsiella pneumoniae*, and *Candida albicans* [27]. It is known that the antimicrobial activity of mucus from terrestrial snails is associated with the presence of AMPs, but also with some antibacterial proteins and glycoproteins. Pitt et al. showed that the mucosa of *H. aspersa* (*Cornu aspersum*) was effective against three laboratory strains of *P. aeruginosa* [30]. In addition, antibacterial proteins and glycoproteins have been reported in the mucus of various terrestrial snails (*A. fulica*, *H. aspersa*, *Cryptozona bistrialis*, *Lissachatina fulica*, and *Hemiplecta differenta*) [18,30,31,32,33].

Our previous studies described the antimicrobial activities of different fractions from the mucus of the garden snail *C. aspersum*, and characterized metabolites and antimicrobial peptides [34,35,36,37]. This study upgraded the previous data and provided additional information on new peptides and protein fractions with antibacterial activity in the mucus of *C. aspersum*. Furthermore, it shed light on the mechanism of antimicrobial action of bioactive compounds from the mucus.

One of the paths to discovery of therapeutic tools and antimicrobial agents with a natural origin that are close to the chemical structure of natural compounds is to use peptides with an antimicrobial action [34,35,36,37]. Peptides with a different molecular weight are known, including peptides and protein fractions with a studied antimicrobial action. Such peptides with antibacterial and antifungal action have been studied previously. A large part of the research was conducted using peptides and protein fractions isolated from snail slime. The present investigation contributed to the efforts to find new natural compounds that can serve as antimicrobial agents with less toxicity for humans and avoid antimicrobial resistance. It also explored the possibilities for minimization of the resources taken from nature, in accordance with some of the main features of bacterial infections.

## 2. Materials and Methods

### 2.1. Mucus Collection and Separation of Different Fractions

The mucus was collected and purified from *C. aspersum* snails grown on Bulgarian farms using a special device with low-voltage electrical stimulation and a method described in BG Utility model 2097/2015 [34,35,36,37]. After removing the coarse impurities from the obtained extract by homogenization and centrifugation, the supernatant was subjected to several cycles of filtration at 4 °C.

The obtained crude mucus extract was divided into two basic fractions by ultrafiltration, using membranes with pore sizes of 10 kDa and 20 kDa (EMD Millipore Corporation, Billerica, MA, USA; and polyethersulfone, Microdyn Nadir™ from STERLITECH Corporation, Goleta, CA, USA, respectively). The peptide fraction with an MW below 10 kDa was further divided into three peptide subfractions by using Amicon^®^ Ultra-15 centrifugal units with 3 and 5 kDa membranes in centrifugation (3000× *g*, 4 °C, 30 min). The used of a noninvasive technique—ultrafiltration—ensured that we obtained fractions containing intact compounds. Finally, the obtained mucus extract was divided into 5 fractions with different molecular weights:

Sample 1—fraction containing compounds with MW < 3 kDa;

Sample 2—fraction containing compounds with MW 3–5 kDa;

Sample 3—fraction containing compounds with MW 5–10 kDa;

Sample 4—fraction containing compounds with MW < 10 kDa;

Sample 5—fraction containing compounds with MW < 20 kDa.

### 2.2. Analysis of Peptide Fractions by Mass Spectrometric Analysis

Peptide fractions with MW < 3 kDa and with MW < 10 kDa were analyzed by MALDI-TOF-TOF mass spectrometry using an Autoflex™ III (Bruker Daltonics, Bremen, Germany) with a laser at a frequency of 200 Hz and operating at a wavelength of 355 nm. The analysis was performed after applying 1.0 μL of a mixture of 2.0 μL of the sample and 2.0 μL of a matrix solution (7 mg/mL α-cyano-4-hydroxyquinamic acid, CHCA) in 50% acetonitrile (ACN) containing 0.1% trifluoroacetic acid (TFA), on a target plate with 192 stainless-steel wells. The mass spectrometer was calibrated with a standard mixture of angiotensin I, Glu-1-fibrinopeptide B, and ACTH (1–17); the ACTH and MS/MS spectra were obtained in reflector mode. The amino acid sequences of the peptides were identified by a MALDI-MS/MS assay using precursor ions from the MS assays.

### 2.3. Antibacterial Activity Testing

The experiments were performed by using a reference strain—the Gram-negative bacterial strain *E. coli* NBIMCC 8785 (NBIMCC, Sofia, Bulgaria). It could serve as a model organism for the determination of the changes in bacterial morphology, viability, and metabolic activity. These microorganisms are among the best-studied bacteria, and provide useful information that can be comparable to many other studies on the subject. They are also a member of the *Enterobacteriaceae* family and can be regarded as a model of opportunistic pathogens with high potential for antibiotic resistance.

### 2.4. Nutrient Media and Culture Conditions

The used nutrient media were nutrient broth (liquid medium) (Himedia Laboratories Pvt Ltd., Mumbai, India) and nutrient agar (solid media) (Himedia Laboratories Pvt Ltd., Mumbai, India). The solid media and the well-diffusion method were used for the determination of the antibacterial activity in the cultivation assays. The bacteria for the SEM and fluorescence analysis were cultivated in nutrient broth. The microbial strains were preserved as a lyophilized culture in glucose medium and peptone protector. In prior experiments, they were rehydrated and maintained at slope agar in tubes. Good laboratory practice was followed during the microbiological assays, and standard microbiological equipment and glassware were used.

### 2.5. Studies of Antibacterial Activities Using Different Methods

For determination of the antibacterial activity, the well-diffusion method was applied with two modifications—deep inoculation and surface cultivation. The deep inoculation included mixing the model bacteria with nutrient agar at a temperature below 40 °C. The growth of the bacteria in the agar volume was a model for infections deeper in the skin. The surface cultivation could serve as model for surface infections. It was performed by using a standardized microbial suspension (50 µL with a density of 10^9^ cells/mL) spread on the top of the agar. The resulting sterile zones were calculated as mm^2^/mgProtein/µMol of the sample. For details, see Dolashki et al. [34] (Figure 1).

The incubation of bacterial cells with the peptide fractions was 1 or 6 h, depending on the variants used in the investigation.

The following variants of the inhibitory effect of the peptide fraction with MW < 10 kDa were investigated: (1) 48 h bacterial culture + 50% peptide fraction—1 h incubation; (2) 18 h bacterial culture + 50% peptide fraction—1 h incubation; and (3) 18 h bacterial culture + 1%, 5%, and 10% peptide fraction—6 h incubation.

Cultivation for the investigation of the inhibition of the activity of *E. coli* NBIMCC 8785 was carried out in the nutrient broth liquid medium. The 18 h and 48 h cultures were obtained by means of cultivation at 36 °C in 300 cm^3^ flasks. The obtained microbial material was divided in 30 cm^3^ flasks. Peptide fractions of 1%, 5%, 10%, or 50% *v*/*v* were added to every flask. The concentration of proteins in the peptide fraction was 0.5 mg/mL.

These variants corresponded to different infections treated with a peptide fraction with MW < 10 kDa in different concentrations.

The bacterial cultures were used in different phases of their development. The 18 h culture was at the late exponential phase of growth, when the microorganisms had reached their maximal activity. The 48 h culture was a model for a culture in the stationary phase, when the bacteria were closer to the ones in the real infections.

### 2.6. Electron Microscopic Assays

The scanning electron microscopy (SEM) was performed on samples treated with a series of ethyl alcohol in increasing concentrations, as described in Dolashki et al., 2020 [34]. The analysis was used to study the morphological changes in the bacteria at an individual level as a result of the presence of the active compounds from the *C. aspersum* mucus.

### 2.7. CTC/DAPI Staining and Digital Image Analysis

CTC/DAPI staining: For determination of the share of live and dead cells in the samples and for estimation of their metabolic activity, a fluorescent tetrazolium salt was used —5cyano-2,3-ditolyl tetrazolium chloride (CTC) (Merck KGaA, Darmstadt, Germany). The samples were stained for 45 min with a 5 mM concentration of the dye. For visualization of all bacteria, DAPI (4′,6-diamidino-2-phenylindole) (Merck KGaA, Darmstadt, Germany) staining was applied at a concentration of 1 µM/mL of the indole dye.

The fluorescence images were taken with a Leica DM6 B (Leica Camera AG, Wetzlar, Germany) epifluorescent microscope. The settings of the camera were identical for all pictures taken during the experiment.

Digital image analysis: The fluorescence images from the CTC/DAPI staining were subjected to digital image analysis using the software daime 2.2 (University of Vienna, Vienna, Austria). The threshold criteria for image segmentation were chosen manually. The digital analysis was made using several parameters.

Intensity of the CTC fluorescence—the intensity of the fluorescence emitted by the CTC stained cells corresponded to their level of metabolic activity. The data were obtained by extracting the mean intensity of the microorganisms on a given image. The analysis was made using three to five CTC images taken in a given sample.

Mean area of the cells—using daime, an analysis of the mean area of the cells on the fluorescent images was made. These data provided information on whether the cells had increased or decreased in volume. The latter were morphological changes that could indicate inhibition of the bacteria’s development. The calculations were based on DAPI staining because the cell morphology was more visible.

Perimeter of the cells—in addition to the cells’ mean area, the mean perimeter of the cells on an image was also analyzed. This could show different morphological changes in the bacterial cells, such as changes in their shape. These deformations could be linked to the antibacterial effect of the peptide fractions. The calculations were also based on the DAPI images.

Circularity—the mean circularity of the objects on the images (e.g., bacterial cells) was also analyzed with daime. The maximum value of this parameter is 1 (ideal circle). The lowering of the circularity indicated an elongation of the objects in the images. This parameter allowed the estimation of the morphological changes in the bacterial cells related to inhibition of the mechanisms of separation of the newly formed cells during division.

The share of live bacteria after the application of the antimicrobial peptides was calculated based on the number of CTC-stained cells (live cells) and DAPI (all cells, including dead ones). The calculations were performed on at least 10 fluorescent images for each sample. The share was represented as a percentage of the live cells from the total bacterial cells in the sample.

All investigations were realized in 3–6 repetitions. The data are mean values, and were statistically treated according to Student and Fisher with a guaranteed probability of 95%.

## 3. Results

### 3.1. Analysis and Physicochemical Characteristics of the Fraction

The discovery of new antimicrobial peptides (AMPs) from natural sources is of great public health importance due to the effective antimicrobial activity of AMPs and low levels of resistance. In the present study, five mucus fractions from the garden snail *C. aspersum* (four peptide fractions and one fraction containing peptides and polypeptides with MWs below 20 kDa) were used.

The extraction of crud mucus from the *C. aspersum* snails grown on Bulgarian farms was performed using a special patented technology, without injuring any snails [34,35,36,37]. After a series of purifications, the crude mucus extract, which was a multicomponent mixture of substances with different masses and properties, was divided by ultrafiltration into two basic fractions with molecular weights below 10 kDa and below 20 kDa. To characterize it in more detail, the peptide fraction with a molecular weight below 10 kDa after ultrafiltration by using Amicon^®^ Ultra-15 with 3 and 5 kDa membranes in centrifugation (3000× *g*, 4 °C, 30 min) was further divided into three subfractions with MW < 3 kDa, MW of 3–5 kDa, and MW of 5–10 kDa, as well as a fraction containing peptides and polypeptides with MW < 20 kDa, as described in the Section 2.

The exact molecular weights of the peptides in the fraction with MW < 3 kDa were determined as protonated molecular ions [M + H]^+^ by the MALDI-TOF/MS analysis performed in positive ionization mode. The MS spectrum of the fraction presented in Figure 2 shows that the snail mucus contained various peptides with different masses, primarily in the region between 800–2500 Da.

The fraction with MW < 10 kDa, in addition to the peptides described above, contained peptides with higher molecular weights. They were determined by MALDI-TOF-MS analysis, and the MS spectrum was dominated by peptides represented as protonated molecular ions at *m*/*z* 3912.91 Da, 4041.15 Da, and 5197.26 Da, and various ions with lower intensities of 6194.62–7060.37 Da (Figure 3). In a previous study, the glycosylation screening by orcinol/H_2_SO_4_ showed that some of the mucus fractions were glycosylated [34], which confirmed the presence of glycopeptides.

Information on the amino acid sequences of the peptides with low molecular weights was obtained by MALDI-MS/MS analysis of the protonated molecule ions [M + H]^+^.

As shown in Figure 4, after following of fragment y- and b-ions, the amino acid sequence of a peptide, represented as a molecular ion [M + H]^+^ at *m*/*z* 1396.96 Da, was identified: EPGGGGGEGGGLLGVAL. Using the same method, the primary structures of the peptides presented in Table 1 were determined by de novo sequencing experiments (MS/MS analysis) of the protonated molecule ions [M + H]^+^.

The amino acid sequences (AASs) of the identified peptides in the fraction with MW below 3 kDa from the *C*. *aspersum* mucus showed the presence of various amino acid residues, but mostly glycine (G), leucine (L), valine (V), proline (P), tryptophan (W), glutamic acid (E), aspartic acid (D), phenylalanine (F), and arginine (R), which are typical for peptides with established antimicrobial activities. The physicochemical characteristics of the identified peptides (isoelectric points (pI), grand average of hydropathicity (GRAVY), and net charge) were determined using the ExPASy ProtParam tool. The analysis showed the presence of both cationic and anionic amphipathic peptide structures. Most peptide structures showed generally hydrophobic surfaces (Table 1), but only two peptides were hydrophilic (Nos. 2 and 11).

Based on the identified primary structures (Table 1), antimicrobial activity was predicted using iAMPpred software (http://cabgrid.res.in:8080/amppred) (accessed on 10 January 2022), an extensive database [38]. The iAMPpred software predicted antimicrobial peptides would be incorporated into compositional, physicochemical, and structural features.

The results showed that peptides 9, 11, 12, 13, 15, and 16 had the highest prognostic, antibacterial, and antifungal activities, while peptides 5, 11, and 12 had the highest prognostic antiviral activities.

Peptides 4, 7, and 8 had the lowest prognostic microbial activities. Some previous studies have shown [39] that if Pro residues are inserted into the sequences of α-helical AMPs, their ability to permeabilize the bacterial cytoplasmic membrane decreases substantially along with the number of Pro residues incorporated, which could explain our results. Eight peptides of the identified sequences contained one to three Pro amino acid residues in the polypeptide chain. Pro residues were incorporated in the N-terminal region of the polypeptide chain in four peptide structures, and in two peptides Pro residues were located in the C-terminal. Only one peptide contained proline residues both in the N-terminal and C-terminal polypeptide chains. Moreover, one proline residue was found in the middle of the polypeptide chain for four peptides (Nos. 4, 7, 8, and 14). Although Pro residue is commonly known as an α-helix breaker, proline residues have been found in the alpha-helical regions of many peptides and proteins, as well as AMPs. Confirmation of these data was provided by the identified structure of peptide No. 16, which contained two Pro residues located at the C-terminus, and showed high prognostic antibacterial and antifungal activities—78.6% and 61.5%, respectively.

The alignment of AASs of the mucus peptides of the garden snail *C. aspersum* presented in Table 1 was acquired with CAMPSing software (http://www.campsign.bicnirrh.res.in/blast.php) (accessed on 10 January 2022), and revealed high homology of peptides 1, 6, 9, 11, 12, 13, 15, and 16 with known AMPs such as microcin, leptoglycin, glycine-rich protein GWK from *Cucumis melo*, ctenidin-1, ctenidin-3, holotricin-3, acanthoscurrin-1 (presented in Appendix A), which confirmed that they belong to the AMP family. The obtained results can be considered as basic information in the study of bioactive peptides from the *C. aspersum* mucus extract and for their potential biomedical application.

### 3.2. The Antibacterial Activity—24 h Impact on Cells of E. coli in a Late Logarithmic Phase in a Solid Nutrient Media with Deep and Surface Inoculations of the Bacterial Material

The antibacterial effect of the peptide and protein fraction regarding *E. coli* NBIMCC 8785 was studied using surface and deep inoculations of the bacterial material. It was registered that the peptide fraction with MW < 10 kDa manifested an antibacterial effect in the deep and surface inoculations. This indicated that this reaction could be used for the treatment of infections that are based on the formation of biofilms, and as such, that are in the interior of the organs and liquids. It would be active in the studied bacterial cultures for bacterial cells immobilized on surfaces, as well as for the homogenous cells gravitating around the biofilm that reproduce the biofilm and release metabolic products of infections with toxic effects [5,40,41,42,43,44,45].

The antibacterial effects regarding *E. coli* NBIMCC 8785 in a deep and surface inoculations of different fractions of peptides and proteins isolated from snail slime is represented in Figure 5. The results showed that the peptide fraction of MW < 10 kDa had an effect on both types of inoculation. This fraction also included the peptides with an MW < 3 kDa, an MW of 3–5 kDa, and an MW of 5–10 kDa.

This fraction includes both peptides with MW < 3 kDa (Figure 2) and the peptides of higher molecular weights shown in the MS spectrum in Figure 3.

The activity in the surface inoculation sample reached a high value of 1990 mm^2^/mgPr/µMol sample, and in the deep inoculation sample, it reached 136.13 mm^2^/mgPr/µMol. This fraction could be applied in the treatment of biofilm of *E. coli* NBIMCC 8785, as shown by the data from the surface inoculation, but the inhibiting effect also would be distributed over the homogenous bacterial cells, which usually gravitate around the biofilm or are only on the homogenous cells [5,40,41,42,43,44,45].

The fraction with MW < 20 kDa had an inhibiting action, reaching 7760 mm^2^/mgPr/µMol for *E. coli* NBIMCC 8785, but only in the surface inoculation of the bacterial material and in the nutrient media.

Our future studies will continue with this peptide fraction, aiming to demonstrate some of the mechanisms of action of the peptides in it on the bacterial cells of *E. coli*. Different experimental variants were studied, corresponding to different scenarios/algorithms for therapeutic applications. The variants chosen by us for the interaction of antimicrobial peptides and bacterial cells were consistent with the special features to be used in future therapeutic processes that will further our study of the mechanisms of practical applications of the characterized peptide fractions.

### 3.3. Antibacterial Activity—First Variant

The impact of the peptide fraction MW < 10 kDa at a high concentration of 50% (0.5 mg/mL protein) on a 48 h bacterial culture with an exposition of action of 1 h was determined. The experiments were conducted in a liquid medium. In terms of therapeutic impact, this meant infections were in an advanced phase, when the treated has area contained bacteria of different ages that formed a bacterial succession. Another specificity for such pathogenic cultures is that there are also metabolic molecules around them from the dying bacterial cells. For the treatment of such infections, usually high initial concentrations of the therapeutic agent are used for the fast and effective manifesting of the bactericidal and bacteriostatic effects. The specific algorithm of the impact of the bacterial cells used was: high concentration of the peptide 50% *v*/*v* (0.5 mg/mL protein) for a short time—1 h impact on a 48 h bacterial culture of *E. coli* NBIMCC 8785.

The effects were determined through the study of the changes in the morphology of the bacterial cells using a scanning electronic microscope, as shown in Figure 6.

In Figure 6, it can be seen that massive damage to the surface cell layers occurred and an unformed cell-free bacterial mass was created that was unable to reproduce bacteria; and at the large scale, there was a limitation of the infection [46,47,48,49].

### 3.4. Antibacterial Activity—Second Variant

The impact of the peptide fraction MW < 10 kDa (0.5 mg/mL protein) at a high concentration of 50% (0.5 mg/mL protein) on 18 h bacterial culture with an exposition of impact of 1 h was investigated. The experiments were conducted in a liquid medium. In this course of experiments, we shaped the therapeutic situation in which the peptide fraction MW < 10 kDa treated infections that had arisen before 18 h. The algorithm was:

High concentration of the peptide 50% (0.5 mg/mL protein) for a short time—1 h of impact in an 18 h bacterial culture of *E. coli* NBIMCC 8785.

The analysis of the mechanisms of impact in this course of experiments was done by determining the level of damage of the metabolism of the bacterial cell through a dyeing of the *E. coli* NBIMCC 8785 cells impacted by the peptide fraction with the fluorescent tetrazolium salt 5-cyano-2,3-ditolyl tetrazolium chloride (CTC).

Figure 7 shows a bacterial culture at 18 h with a density of 1.8 × 10^7^ cell/mL with 50% peptide fraction after dyeing with CTC. It can be clearly seen that in the control, there was a large number of metabolically active bacterial cells of *E. coli* NBIMCC 8785. After the one-hour impact on the bacterial culture with a peptide fraction MW < 10 kDa, a strong inhibiting effect was seen in the metabolism of the bacterial agent [9,50,51].

The fluorescent images of the control and of the bacterial cells impacted by the peptide fraction MW < 10 kDa were digitally processed. The digital processing was performed on the following indicators: fluorescence intensity, average perimeter of the cells, roundness, and average area of the cells. The results of the changes to these parameters of the bacterial cells impacted by the peptide fraction MW < 10 kDa regarding the control are represented in Figure 8.

The results indicated that the three parameters decreased after the impact by the peptide fraction MW < 10 kDa. This meant that the fluorescence intensity decreased by about 20%, the average perimeter decreased by about 15%, the cells’ length and roundness decreased by 4%, and the average area decreased by about 35%. Using the represented fluorescent images and their digital analysis, it was found that after the 1 h impact with the peptide fraction MW < 10 kDa, the bacterial cells had inhibited functions and had changed morphology. On the other hand, using Figure 9, it was determined that at this concentration, for the 18 h culture and one hour of exposition, there was a very high level of inhibition of the bacterial cells. As a total result of the registered cell deformations and the fluorescence intensity, there was almost 100% inhibition of the structures and functions of the pathogenic bacterial cells. It was determined that the penetration of peptides with low molecular weights from the outside to the inside of bacterial cells damaged the cells’ surface layers, thereby inhibiting the metabolic activity of the bacteria. Various morphological deformations of the cells were registered that were illustrated by the SEM analyses—folding of the surface cell layers, and lengthened and shortened cells [43,46,47,48,49,52].

### 3.5. Antibacterial Activity—Third Variant

Logic led us to question whether it was possible for the initial infection to have a decreased concentration of the therapeutic agent in the peptide fraction MW < 10 kDa (0.5 mg/mL protein) while, at the same time, having an increased exposition time. This algorithm for the treatment of the bacterial cells with the peptide fraction with MW < 10 kDa with the increase in the exposure time would lead to a series of positive effects: the use of a smaller quantity of peptide fractions that would be favorable regarding costs, emanation of therapeutic agents into the environment, and economization of resources at the expense of the exposition time. The studied algorithm was:

Impact of the peptide fraction MW < 10 kDa (0.5 mg/mL protein) with concentrations 1, 5, and 10% on 18 h bacterial culture with an exposition of 6 h.

The experiments were conducted in a liquid medium. Through this course of experiments, we shaped the therapeutic situation in which relatively newly formed infections (arisen before 18 h) were treated with the peptide fraction MW < 10 kDa. The aim was to choose the convenient, comparably low concentration of peptide fraction. The analysis of the mechanisms of impact in this course of experiments has been made using the level of damage of the metabolism of the bacterial cells through the dyeing of the *E. coli* NBIMCC 8785 cells impacted by the peptide fraction with the fluorescent tetrazolium salt 5-cyano-2,3-ditolyl tetrazolium chloride (CTC) and through SEM.

The fluorescent images of the obtained results are presented in Figure 9. In the presented images, it can be clearly seen that with the increase in the concentration of the peptide fraction from 1% to 10%, the number of dead inactive bacterial cells also increased. The highest effectiveness of inhibiting actions was realized using the concentration of the peptide fraction of 10%. After that, we also conducted a digital analysis of the images using the following parameters: fluorescence intensity, average perimeter of the bacterial cells, roundness, average area of the cells, and ratio of alive to dead cells.

The results shown in Figure 10 provided grounds to reach the following intermediate conclusions. In the 1% peptide fraction, only morphological changes were registered in the cells of *E. coli* NBIMCC 8785, represented by the lengthening and decrease in the average area of the cells. Regarding the impact of the 5% peptide fraction, the three morphological parameters of the cells decreased—the average perimeter and average area decreased and the cells lengthened, but the metabolic activity was not affected (See Appendix A).

In the 10% (*v*/*v*) peptide fractions with 0.5 mg/mL protein and 6 h of incubation, a high inhibiting effect of the peptide fraction regarding *E. coli* NBIMCC 8785 was registered—a strong decrease in the fluorescence intensity meant a decrease in the metabolic activity, as well as an increase in the average perimeter and the average area of the cells, and their lengthening.

After that, the mechanisms of damaging of the form and sizes of the bacterial cells in this algorithm of impact were confirmed by the SEM analyses. When comparing the effects of the damage to the cells of *E. coli* NBIMCC 8785 with the peptide fraction MW < 10 kDa at a concentration of 10% and 6 h of exposition, the following results were found. Morphologically, the bacterial cells were damaged, and the following deformations were registered: (1) strong lengthening of the cells—the lengthened cells reached three times the size of the normal cells (Figure 11a,d); (2) Cell cracking and pouring of the cells’ contents into the outside cell space (Figure 11a,c); (3) shortening of the cells and pinching in the middle, folding, and concavities on the cells’ surfaces (Figure 11a,b,d,f); and (4) conglutination of the cells and the obtaining of an unformed bacterial cell mass (Figure 11a,d,f).

One of the most unexpected effects of damage of bacterial cells registered by SEM was a triple increase in the size of the cells.

The strongly lengthened cells also were registered through a fluorescent technique—a dyeing with the fluorescent labelled tetrazolium salt 5-cyano-2,3-ditolyl tetrazolium chloride (CTC). The results illustrating these cells that were three times larger that were obtained after the impact of the peptide fraction MW < 10 kDa for 6 h are presented in Figure 12a,b.

Comparing the data in Figure 12a,b, we supposed that one of the effects of damage was the blocking of the division of the cells, which impeded their multiplication. Thus, the results at the cell level showed these long cell structures. This phenomenon registered and confirmed by us was accompanied by the reported effects of damage to the existing cells, leading to a fast termination of the infections that would not allow their development. In this way, the early infections were limited and terminated at a low concentration of the peptide fraction in a longer time of exposition.

## 4. Discussion

In recent years, the mucus from the garden snail *C. aspersum* has been the subject of research by our scientific team. Using tandem mass spectrometry, the primary structures of nine antimicrobial peptides with molecular weights of 1000–3000 Da in an isolated fraction with MW < 10 kDa from mucus from the garden snail *C. aspersum* with active components were identified. These peptides were rich in glycine and leucine residues, and demonstrated strong antibacterial activity against a Gram-negative bacterial strain—*E. coli* NBIMCC 878 [36]. Moreover, some peptides with MW 10–30 kDa exhibited predominantly antibacterial activity against *B. laterosporus* and *E. coli*, and another 20 kDa peptide fraction against the bacterial strain *C. perfringens* [34].

A characteristic of the structures of peptides with the highest prognostic, antibacterial, and antifungal activity was the content of high levels of glycine and leucine residues, which indicated that they belonged to a new class of antimicrobial peptides rich in Gly/Leu that demonstrated antibacterial activity mainly against Gram-negative bacteria [53]. Previous studies also showed the presence of peptides with similar amino acid sequences [34,36,37]. Furthermore, four of the nine peptides with high prognostic therapeutic values were cationic, and only one was neutral. It is known that cationic AMPs kill microbes via mechanisms that predominantly involve interactions between the peptide’s positively charged residues and anionic components of the target cell’s membranes. These interactions can lead to a range of effects, including membrane permeabilization, depolarization, leakage, or lysis, resulting in cell death. Some of the positively charged AMPs may penetrate into the cell to bind intracellular molecules that are crucial to the life of the cell. There are multiple models of mechanisms to explain the action of these peptides, including the toroidal pore model, the barrel-stave model, and the carpet model [54,55,56].

Our previously published data in Dolashki et al., 2020 showed that some peptides that were detected by an orcinol/H_2_SO_4_ assay were glycosylated [34]. The glycosylation of peptides can influence their antimicrobial activity and their ability to affect host immunity, target specificity, and biological stability [57,58,59,60]. Glycosylation of AMPs does not necessarily result in the generation of an efficacious peptide, and can sometimes lead to a loss in activity or functionality [61].

The observed antibacterial activity against *E. coli* NBIMCC 8785 most likely can be attributed to the peptides presented in Table 1 (Nos. 1, 11–13, 15, and 16) for which the iAMPpred software predicted high antimicrobial activity, and for which high homology with known AMPs with activity against Gram-negative bacteria was detected, such as leptoglycin, microcin, ctenidin, and acanthoscurrin (See Appendix A). However, based on previous research [34,62], our hypothesis was that the antibacterial activity of the mucus fraction < 10 kDa was due to a synergistic effect of cationic, anionic, and neutral peptides with MWs of 0.800–2.500 kDa and peptides with a higher MW in the range of 3–10 kDa (Figure 3); some of them were glycopeptides, as reported previously in [34]. In fact, Trapella et al., 2018, also suggested that snail mucus’s potential (HelixComplex) as a therapeutic agent in wound repair were attributable to the synergistic activity of several molecules [63].

The results of our study indicated the great opportunities for the isolated and characterized peptide fractions of snail mucus to be used for treatment of infections caused by *E. coli* NBIMCC 8785. The results of the conventional study of the antibacterial activity of the fractions with MW < 3 kDa, MW 3–5 kDa, MW 5–10 kDa, MW < 10 kDa, and MW < 20 kDa indicated that the fraction of peptides MW < 10 kDa had a higher antibacterial effect in surface and deep inoculations of the bacterial material [34,36,37,57]. This fraction would be effective in the treatment of infections that form a bacterial biofilm, and it would impact on the homogenous cells that gravitate around and support the biofilm. On the other hand, this peptide fraction will be effective on the studied bacteria in surface infections and in infections situated deeper.

The peptide fraction with MW < 10 kDa would be an interesting subject of future studies of the mechanism of the antibacterial activity due to its specific actions against *E. coli* and its composition clarified from a chemical point of view. Other factors that make this fraction convenient for application are, on one hand, the lower molecular weight of the peptides means they can more easily penetrate into the bacterial cells; and on the other hand, the surplus of these therapeutic tools will degrade faster in the environment [9,10,43,50].

Depending on the type and the phase of development of the infection, different concentrations and quantities of the peptide fraction could be used. In infections in an advanced-phase 48 h bacterial culture, it was effective to apply a 50% peptide fraction. Using this concentration for a short time of 1 h, a strong inhibiting effect on the bacterial cells was registered, represented by major damage to the cells’ surface layers [5,9,47,48,49,50,52,64]. In the bacterial infections in an initial phase (18 h bacterial culture), a 50% peptide fraction with one-hour impact lead to an almost 100% inhibition of the structures and functions of the *E. coli* NBIMCC 8785. The morphological parameters of the cell were changed, and the metabolic activity was inhibited at a very high level. When studying the opportunities for the achievement of the desired effect in newly arisen infections through a decrease in the concentration of the peptide fraction but with an increase in the exposition time, we found that the 10% peptide fraction led to the achievement of the desired effect—damage to the structures and functions of the bacteria [65,66,67]. This led to a decrease in the fluorescence as a result of the blocking of the oxide reductase apparatus of the cells, and accumulation of degenerative bacterial cells, which were shortened and round, and had highly damaged surfaces. Strongly lengthened cells were registered due to the suppression of their division and the multiplication of new bacterial cells. Similar effects were seen by other authors [43,46,47,48,49,51].

In conclusion, this paper presented 14 new antimicrobial peptides with MWs below 10 kDa isolated from the mucus of *C. aspersum* snails. Their AAS, physicochemical characteristics, and potential antibacterial activities were determined by de novo sequencing. Information on the mechanisms of antibacterial activities of peptides against *E. coli* NBIMCC was presented for the first time by determining the effective concentration and methods of administration of the active fraction with MW < 10 kDa.

Changes in the bacterial structure and metabolic activity of *E. coli* NBIMCC were determined by SEM, fluorescence, and digital image analysis.

These studies shed new light on the mechanisms of action of the peptide fraction with MW < 10 kDa, and allowed us to offer an objective algorithm for the treatment of infections, depending on the stage of development of the bacterial infection [13,68,69,70]. Reducing therapeutic concentrations will be more environmentally friendly and will reduce the cost of therapeutic procedures.

## Figures and Tables

**Figure 1 biomedicines-10-00672-f001:**
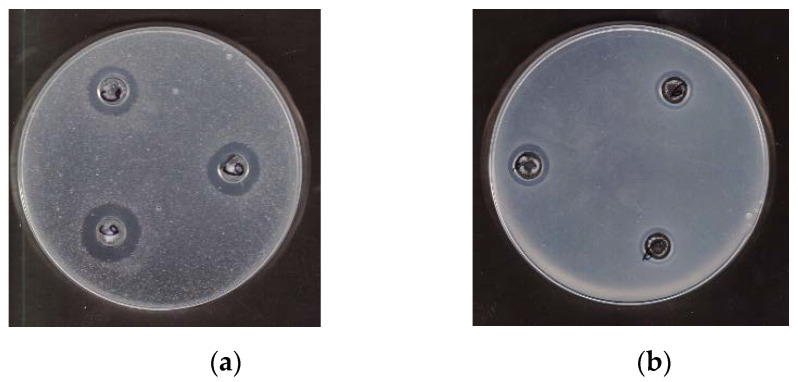
Images of sterile areas found in the investigation of antibacterial activity against model bacteria: antibacterial effect of peptide fraction MW < 10 kDa on *E. coli* NBIMCC 8785 at surface cultivation (**a**) and at inoculation in depth (**b**).

**Figure 2 biomedicines-10-00672-f002:**
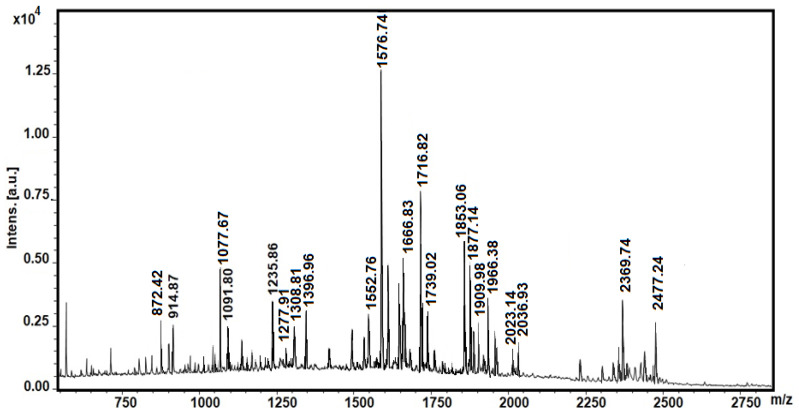
MALDI-MS spectrum of fraction with MW < 3 kDa. Standard peptide solution was used to calibrate the mass scale of the Autoflex™ III High-Performance MALDI-TOF and TOF/TOF Systems (Bruker Daltonics, Bremen, Germany).

**Figure 3 biomedicines-10-00672-f003:**
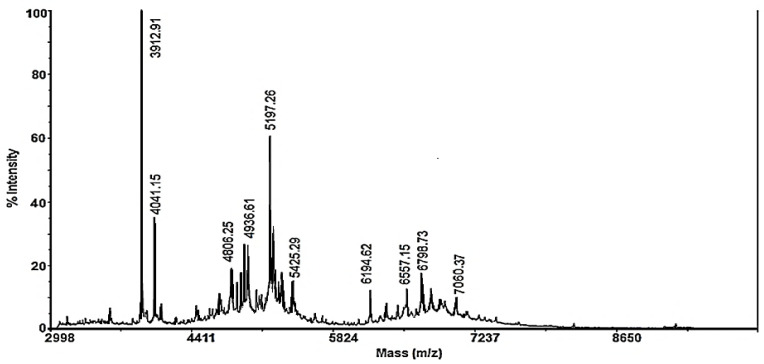
MALDI-MS spectrum recorded in the region of 3000–10,000 *m*/*z* of the fraction with MW < 10 kDa. Standard peptide solution was used to calibrate the mass scale of the Autoflex™ III High-Performance MALDI-TOF and TOF/TOF Systems (Bruker Daltonics, Bremen, Germany).

**Figure 4 biomedicines-10-00672-f004:**
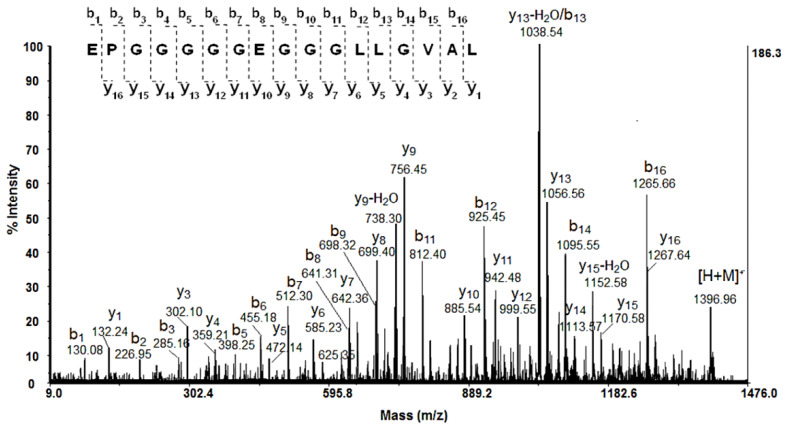
MALDI-MS/MS spectrum of peptide [M + H]^+^ at *m*/*z* 1396.96 Da. The sequence EPGGGGGEGGGLLGVAL was determined by de novo sequencing.

**Figure 5 biomedicines-10-00672-f005:**
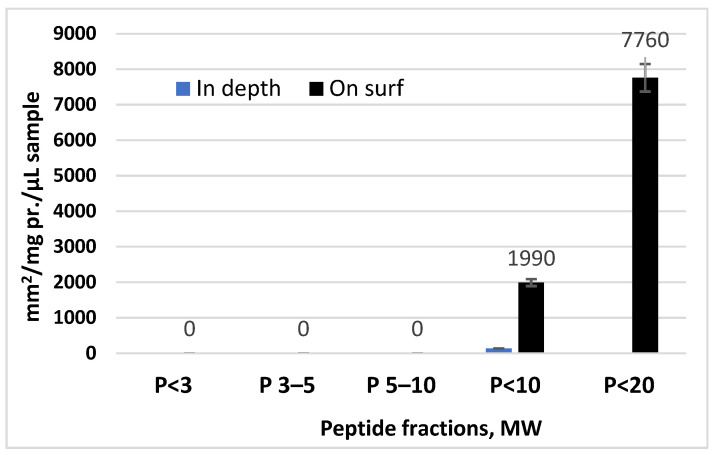
Antibacterial effect of the peptide and protein fractions regarding *E. coli* NBIMCC 8785 in surface and deep inoculations.

**Figure 6 biomedicines-10-00672-f006:**
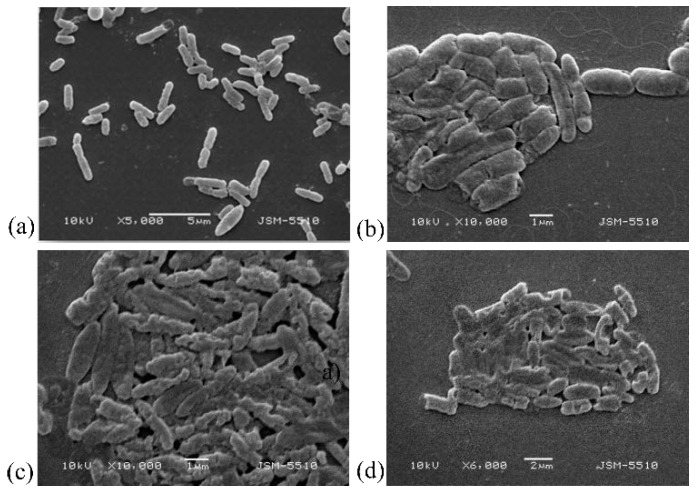
SEM analysis of cells of *E. coli* NBIMCC 8785 influenced with 50% peptide fraction (MW < 10 kDa): (**a**,**b**)—control 48 h bacterial culture at 5000× and 10,000× magnification; (**c**,**d**) damaged bacterial cells of *E. coli* NBIMCC 8785 after action of the peptide fraction.

**Figure 7 biomedicines-10-00672-f007:**
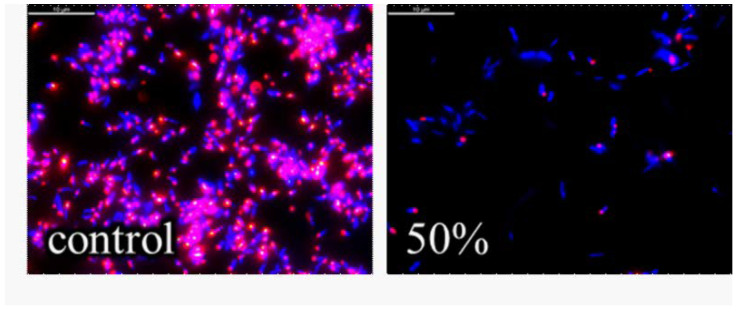
Fluorescence images of *E. coli* after 1 h incubation with peptide fraction with MW below 10 kDa (red—live cells; blue—dead cells). The marker corresponds to 10 µm.

**Figure 8 biomedicines-10-00672-f008:**
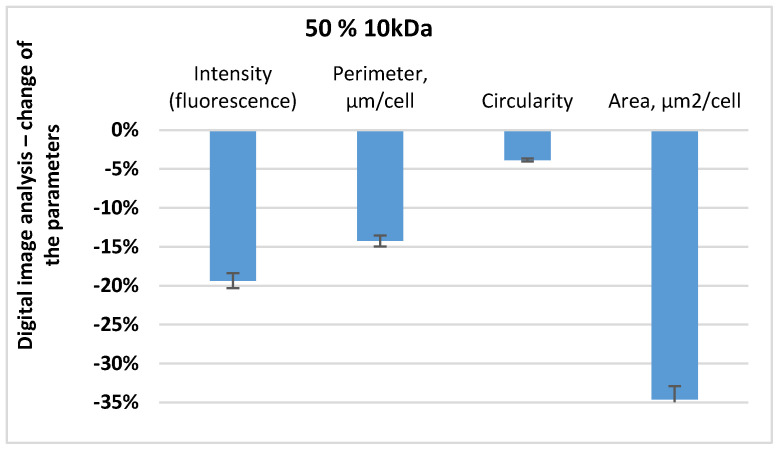
Fluorescence images of *E. coli* after 1 h incubation with peptide fraction with MW below 10 kDa.

**Figure 9 biomedicines-10-00672-f009:**
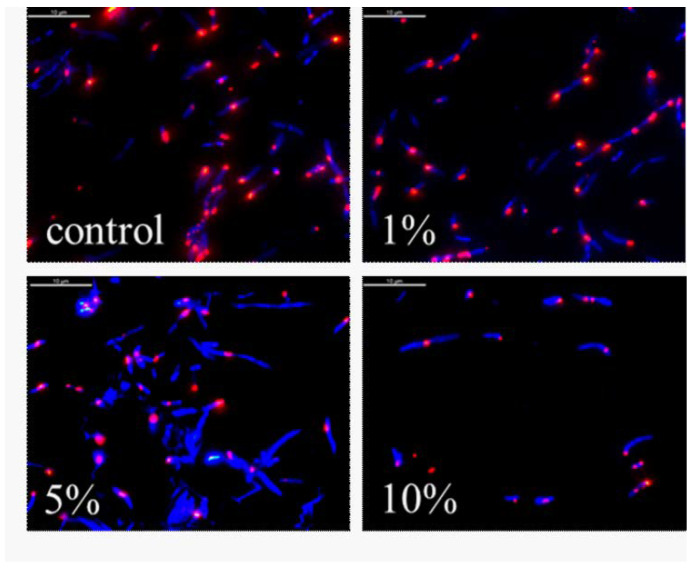
Fluorescence images of *E. coli* after 6 h incubation with peptide fraction with MW below 10 kDa (red—live cells; blue—dead cells). The marker corresponds to 10 µm.

**Figure 10 biomedicines-10-00672-f010:**
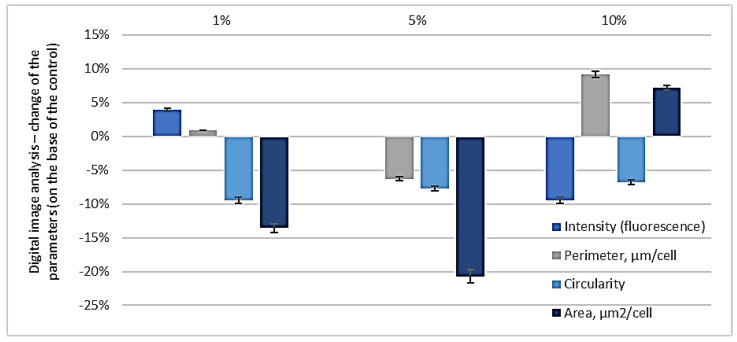
Changes in the fluorescence intensity, parameters of the cells, circularity, and area of *E. coli* cells compared to the control after incubation with different concentrations (1%, 5%, 10%) of the peptide fraction with MW below 10 kDa or 6 h.

**Figure 11 biomedicines-10-00672-f011:**
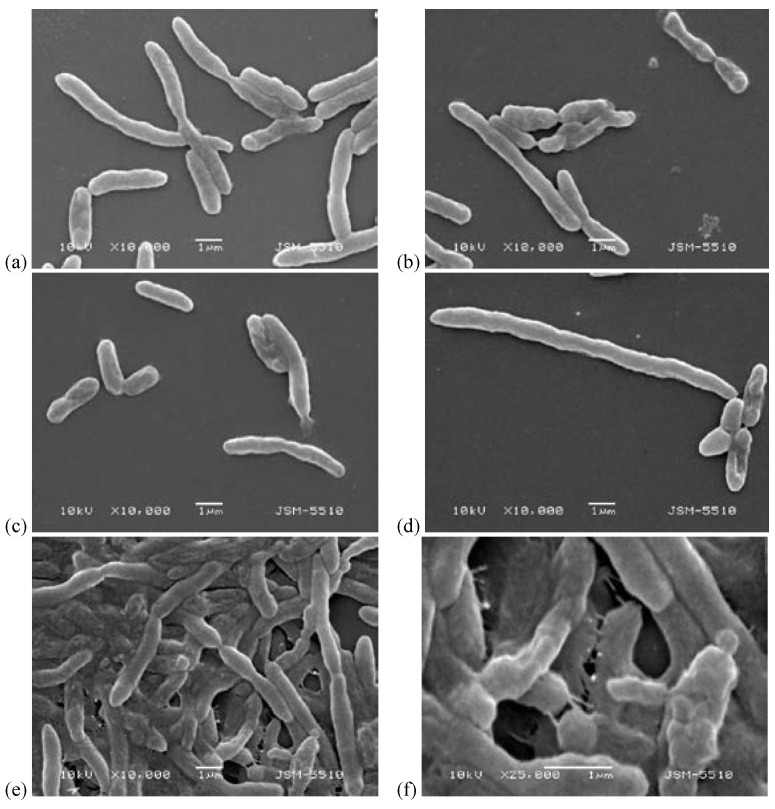
SEM analysis of cells of *E. coli* NBIMCC 8785 influenced with 10% peptide fraction (MW < 10 kDa): (**a**) control—18 h bacterial culture at 10,000× magnification; (**b**–**f**) damaged bacterial cells of *E. coli* NBIMCC 8785 after action of the peptide fraction for 6 h.

**Figure 12 biomedicines-10-00672-f012:**
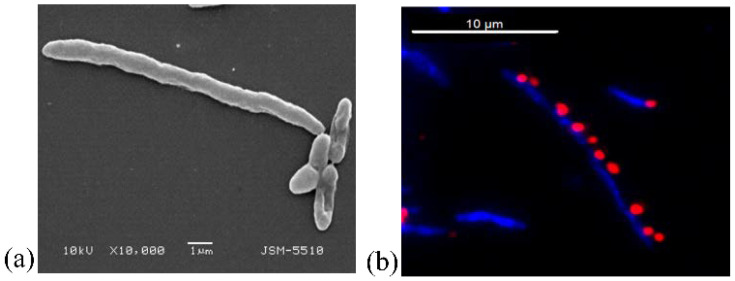
Illustration of long cells (results of blocking of division of cells) of *E. coli* NBIMCC 8785 after 6 h treatment with 10% peptide fraction, MW < 10 kDa: (**a**) SEM analysis at 10,000×; (**b**) fluorescence analysis, with the cells stained with DAPI and CTC.

**Table 1 biomedicines-10-00672-t001:** The primary structures of the peptides presented in mucus fraction below 3 kDa, from the garden snail *C. aspersum*, identified by de novo sequencing in MALDI-MS/MS.

No.	Amino Acid Sequence of Peptides (N-C-Terminals), De Novo Sequencing.	MALDI [M + H]^+^ (Da)	Calc. Mass (Da)	pI	GRAVY	Net Charge	Predict. Antibacterial (%)	Predict. Antiviral (%)	Predict. Antifungal (%)
1	AAGLAGAGGGGGG	872.42	871.41	5.57	0.600	0/0	58	41	32
2	DKGLGGFEA	893.51	892.43	4.37	−0.411	−2/+1	30	29	20
3	LGDLNAEFAAG	1077.67	1076.51	3.67	0.409	−2/0	27	46	30
4	AGVGAGGANPSTYVG	1277.91	1276.60	5.57	0.260	0/0	25	7.5	11
5	GAACNLEDGSCLGV	1308.81	1307.55	3.67	0.564	−2/0	58	58	53
6	EPGGGGGEGGGLLGVAL	1396.96	1395.70	3.80	0.306	−2/0	41	35	20
7	LGPLYDEMGPVGGDVG	1576.74	1574.73	3.49	0.056	−3/0	9.7	20	6.2
8	ASKGCGPGSCPPGDTVAGVG	1716.82	1715.76	5.86	0.005	−1/+1	25	12	23
9	ACSLLLGGGGVGGGKGGGGHAG	1739.02	1737.86	8.27	0.409	0/+1	83	49	67
10	ACLTPVDHFFAGMPCGGGP	1877.14	1875.81	5.08	0.542	−1/0	32	43	20
11	NGLFGGLGGGGHGGGGKGPGEGGG	1909.90	1908.88	6.75	−0.487	−1/+1	90	67	80
12	LLLLMLGGGLVGGLLGGGGKGGG	1966.24	1965.14	8.75	1.209	0/+1	92	57	76
13	PFLLGVGGLLGGSVGGGGGGGGAPL	2023.14	2022.09	5.96	0.912	0/0	69	32	38
14	GMVVKHCSAPLDSFAEFAGA	2036.93	2035.95	5.32	0.565	−2/+1	42	20	26
15 *	GLLGGGGGAGGGGLVGGLLNG	1609.94	1608.86	5.52	0.776	0/+1	90.0	53.6	65.0
16 *	MGGLLGGVNGGGKGGGGPGAP	1666.83	1665.83	8.5	0.005	0/+1	78.6	52.0	61.5

* The AASs of these peptides were also identified previously in [34].

## Data Availability

The data presented in this study are available in the article.

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
