# Peer review of "Effect and Mechanisms of Antibacterial Peptide Fraction from Mucus of C. aspersum against Escherichia coli NBIMCC 8785"

_biomedicines, 2022, doi:10.3390/biomedicines10030672_

Round 1

Reviewer 1 Report

This an interesting paper on the Effect and mechanisms of antibacterial peptide fraction from 2 mucus of Cornu aspersum against Escherichia coli NBIMCC 3 8785. However I would like to draw your attention to the following:

L41 antibiotics are not effective against viruses, hence the correlation is not appropriate. This is a presumptuous statement and should be rephrased or removed.

L118 You are referring to seventeen novel peptides with potential antibacterial activity that were identified by de novo MS/MS sequencing using tandem mass spectrometry ( Dolashki A, et al. Biomedicines. 2020;8(9):315. 2020 doi:10.3390/biomedicines8090315) ? Please point out the difference between the two studies.

intro too extensive on resistance and antimicrobials in general. Should elaborate on snail AMPs and to differences among snail species. Please highlight the novelty of the present study.  

L122 Please provide a brief description of the sample preparation. e.g. you have 4 samples and each of them contains a large number of antibacterials. Are any other proteins or other cell constituents excluded and how? Have you checked antibacterial activity in the isolated fractions to ascertain that they concern antibacterials using Escherichia coli NBIMCC 8785 as the indicator? If so then the part Microbial strains could read Antibacterial activity testing  to make methods more clear

L183 why 430 nm when usually bacterial density is monitored at 600nm? also where are such data presented?

L193 please clarify the phases of growth to which the 48h and 18h samples correspond to. The provision of a  growth curve in the supplement might be sufficient.

L241 please be more specific regarding the statistical tests employed and the reason for being employed.

L 257 Please state the means you employed to verify that all these masses correspond to separate antimicrobial peptides and not fragments of peptides that were created at the process. if this is due to the non-invasive technique employed, please state it. Also did you check antimicrobial action of pooled fractions for each compound?

Table 1 contains a word that is not in English

L377 you are possibly referring to  figure 6 for SEM and 7 for AFM

L 416 why was fluorescence used to evaluate peptide fraction <10 kD?

L450 is it exposition or exposure?

L548 experiments with amylase or lipases could  also clarify the nature of these peptides (glycoproteins, lipoproteins etc) in correlation with their antibacterial action. What is the nature of the peptides used in the present study? Please be more specific regarding the nature of each peptide.

Author Response

DearEditor,

Thank you very much for your deep interest to our paper and for all remarks and comments from Reviewers. We made the necessary improvements and are sending our corrected paper and explanation of your questions.  

Reviewer 1

This an interesting paper on the Effect and mechanisms of antibacterial peptide fraction from 2 mucus of Cornu aspersum against Escherichia coli NBIMCC 3 8785. However, I would like to draw your attention to the following:

Reviewer: L41 antibiotics are not effective against viruses, hence the correlation is not appropriate. This is a presumptuous statement and should be rephrased or removed.

Answer: The sentence The actuality of this requirement increases due to the wider application of antibiotics in the contemporaneity after the peak of distribution of SARS-Cov-2” is removed.

 Reviewer: L118 You are referring to seventeen novel peptides with potential antibacterial activity that were identified by de novo MS/MS sequencing using tandem mass spectrometry (Dolashki A, et al. Biomedicines. 2020;8(9):315. 2020 doi:10.3390/biomedicines8090315)? Please point out the difference between the two studies.

Answer: The presented new study is a continuation of the previous study of Dolashki A, et al., 2020, where 17 peptides were identified. In addition, in this article 14 new peptides were identified and characterized, their amino acid sequences (AAS), as well as their potential antibacterial activity predicted using iAMPpred software. Also the AASs of two peptides at m/z 1609.94 and m/z 1666.83 were confirmed. In this study, new information was presented on the distribution of peptides in a fraction with a molecular weight (MW) below 10 kDa, by MALDI-MS analysis of peptides with MW between 3-10 kDa.

The presented study focuses on the antibacterial activity of the fraction with MW below 10 kDa. For the first time, information on the mechanisms of antibacterial action of peptides against Escherichia coli NBIMCC, the effective concentration and methods of administration of the active fraction was presented. Changes in the bacterial structure and metabolic activity of E. coli NBIMCC were determined by SEM, fluorescence and digital image analysis.

Reviewer: Intro too extensive on resistance and antimicrobials in general. Should elaborate on snail AMPs and to differences among snail species.

Answer: The introduction of resistance and antimicrobials is shortened. Information has been added on the antimicrobial activity of mucus from various terrestrial snails, which has been linked to the presence of peptides but also to proteins and glycoproteins.

Line 91-109- For example, the garden snail (Cornu aspersum) and the giant African snail (Achatina fulica) are amongst the most studied land snails [18, 27]. The water gastropods also offer a variety of AMP. Some of the biologically active molecules identified are developed in pharmaceuticals like Prialt®, Adcetris®, and many others [28].

Several peptides from the hemolymph of garden snails H. lucorum and H. aspersa have been reported to show a broad spectrum of antimicrobial activity against Staphylococcus aureus, Staphylococcus epidermidis, E. coli, Helicobacter pylori and Propionibacterium acnes [22, 26]. The mucus content of snails varies depending on the function and secretory structure of the glands and species of gastropods [30]. Zhong et al., found AMP "Mytimacin-AF" (9700 Da) in Achatina fulica mucous, which exhibits antimicrobial activity against S. aureus, Bacillis spp., Klebsiella pneumoniae and Candida albicans [27]. It is known that the antimicrobial activity of mucus from terrestrial snails is associated with the presence of AMPs, but also with some antibacterial proteins and glycoproteins. Pitt et al. show that the mucosa of H. aspersa (Cornu aspersum) is effective against three laboratory strains of P. aeruginosa [31]. In addition, antibacterial proteins and glycoproteins have been reported in the mucus of various terrestrial snails (A. fulica, H. aspersa, Cryptozona bistrialis, Lissachatina fulica and Hemiplecta differenta) [18, 31-34].

Reviewer: Please highlight the novelty of the present study

Answer: The highlights of the present study are presented in the conclusion:

Line 595-607 - In conclusion, this paper presents 14 new antimicrobial peptides with MT below 10 kDa isolated from the mucus of snails C. aspersum. AAS, their physicochemical characteristics, and their potential antibacterial activity were determined by de novo sequencing. For the first time, information on the mechanisms of antibacterial activity of peptides against E. coli NBIMCC was presented, determining the effective concentration and methods of administration of the active fraction with MT <10 kDa. Changes in the bacterial structure and metabolic activity of E. coli NBIMCC were presented by SEM, fluorescence and digital image analysis.

These studies shed new light on the mechanisms of action of the peptide fraction with MW <10 kDa and allow to offer an objective algorithm for the treatment of infections depending on the stage of development of bacterial infection [13, 70-72]. Reducing therapeutic concentrations will be more environmentally friendly and will reduce the cost of therapeutic procedures.

Lines 121-125 - Additionally, a sentence focused on the contribution of the study was added at the end of the Introduction:

„The present investigation contributes to the efforts for finding new natural compounds that can serve as antimicrobial agents with less toxicity for humans and avoid antimicrobial resistance. It also explores the possibilities for minimization of the resources taken from nature in accordance with some of the main features of bacterial infections.

Reviewer:  L122 Please provide a brief description of the sample preparation. e.g. you have 4 samples and each of them contains a large number of antibacterials. Are any other proteins or other cell constituents excluded and how?

Answer: A brief description of the sample preparation is added in Results.

Lines 254-267 - In the present study, 5 mucus fractions from the garden snail C. aspersum (4 peptide fractions and 1 fraction containing peptides and polypeptides with MW below 20 kDa) were used.

The extraction of crud mucus from the snail C. aspersum grown on Bulgarian farms was obtained using a special patented technology, without damaging any snails [35-38]. After a series of purifications, the crude mucus extract, which is a multicomponent mixture of substances with different masses and properties, was divided by ultrafiltration into two basic fractions with molecular weights below 10 kDa and below 20 kDa. To characterize in more detail, the peptide fraction with a molecular weight below 10 kDa after ultrafiltration with Amicon® Ultra-15 with 3 and 5 kDa membranes, by centrifugation (3000 xg, 4oC, 30 minutes) was further divided into 3 subfactions with MW <3 kDa, with MW between 3-5 kDa and a fraction with MW between 5-10 kDa and a fraction containing peptides and polypeptides with MW <20 kDa) as described in materials and methods.

Reviewer:  Have you checked antibacterial activity in the isolated fractions to ascertain that they concern antibacterials using Escherichia coli NBIMCC 8785 as the indicator? If so then the part Microbial strains could read Antibacterial activity testing to make methods more clear.

Answer: The heading of the section was corrected. The antibacterial activity was checked, as described in Materials and Methods. The text was changed to be clearer.

Lines 157-164: 2.4. Antibacterial activity testing  - „The experiments were performed by using a reference strain – the gram-negative bacterial strain Escherichia coli NBIMCC 8785. It could serve as a model organism for the determination of the changes in bacterial morphology, viability, and metabolic activity. These microorganisms are the best studied and provide useful information that can be comparable to many other studies on the subject. They are also a member of the Enterobacteriaceae family and can be regarded as a model of opportunistic pathogens with high potential for antibiotic resistance“

Reviewer:  L183 why 430 nm when usually bacterial density is monitored at 600nm? also where are such data presented?

Answer: The method is normally used to monitor the accumulation of the biomass during experiments with pure cultures. Bacterial biomass can be monitored in the range 400-600 nm, and for these experiments 430 nm gave the best results and this is founded experimentally. The text was removed because the data wasn’t included in the manuscript.

Reviewer:  L193 please clarify the phases of growth to which the 48h and 18h samples correspond to. The provision of a growth curve in the supplement might be sufficient.

Answer: An explanation was added in the text.

Lines 195-205: „The 18-hours and 48-hours cultures were obtained by means of cultivation at 36oC in 300 cm3 flasks. The obtained microbial material was divided in flasks 30 cm3. To every flask were added peptide fractions with 1%, 5%, 10%, or 50% v/v. The concentration of protein in peptide fraction was 0.5 mg/mL These variants corresponded to different infections, treated with a peptide fraction with MW < 10 kDa in different concentrations. The bacterial cultures are used in different phases of their development. The 18-hour culture was at the late exponential phase of growth when the microorganisms have reached their maximal activity. The-48-hour culture was model for a culture in the stationary phase when the bacteria were more close to the ones in the real infections.“

Reviewer:  L241 please be more specific regarding the statistical tests employed and the reason for being employed.

Answer: The text was corrected:

Lines 245-247: “The calculations were performed on at least 10 fluorescent images for each sample. The share was represented as a percentage of the live cells from the total bacterial cells in the sample.

All investigations were realized in 3-6 repetitions. The data are mean values and were statistically treated according to Student and Fisher with guarantee probability of 95%.”

Reviewer:  L 257 Please state the means you employed to verify that all these masses correspond to separate antimicrobial peptides and not fragments of peptides that were created at the process. If this is due to the non-invasive technique employed, please state it.

Answer: As stated above, to obtain peptide fraction was used a non-invasive technique – ultrafiltration, which ensures production of intact peptides. The purified mucus extracts stored at 4 °C, as previously reported Trapella et al., 2018. An appropriate addition is made in the text.

Reviewer:  Also, did you check antimicrobial action of pooled fractions for each compound?

Answer: As described above, the design of the study included testing of different concentrations of a fraction isolated from snail mucus. The separation was performed by molecular weight and the fraction contained different types of molecules. We haven’t performed antibacterial testing of specific compounds isolated from it.

Reviewer:  Table 1 contains a word that is not in English.

Answer: This error has been corrected.

Reviewer:  L377 you are possibly referring to figure 6 for SEM and 7 for AFM.

Answer: It was corrected in the text.

Reviewer:  L 416 why was fluorescence used to evaluate peptide fraction <10 kD?

Answer: The fluorescence staining was used for evaluation of the bacterial viability and metabolic activity with and without the biologically active compounds from the snail mucus. The peptide fraction wasn’t studied with the fluorescent techniques.

Reviewer:  L450 is it exposition or exposure?

Answer: We meant exposure. it was corrected in the text (line 451).

Reviewer:  L548 experiments with amylase or lipases could also clarify the nature of these peptides (glycoproteins, lipoproteins etc) in correlation with their antibacterial action. What is the nature of the peptides used in the present study? Please be more specific regarding the nature of each peptide.

Answer: Previous studies showed that the snail mucus contained complex mixture of proteoglycans, glycosaminoglycans, glycoproteins, hyarulonic acid, small peptides, low concertation allantoin and glycolic acids, and metal ions [Smith et al., 2009, Trapella et al., 2018; Matusiewicz et al., 2018]. There are no data on the presence of lipoproteins in the mucus from C. aspersum. Recently Matusiewicz et al., reported that the fatty acid profile of H. aspersa Müller mucus was not determined because of the very low fat content [Matusiewicz et al., 2018]. Based on all facts of the above, the presence of lipopeptides is very unlikely, so treatment with amylase or lipases is not necessary.

Based on previous our studies and the results shown in this study, we can conclude that the active fraction with MW below 10 kDa contains mainly peptides and glycopeptides. The peptides presented in Table 1 are intact peptides. Some of the peptides defined as [M + H]+ of the MALDI-MS spectrum are in fact glycopeptides. The results from performed glycosylations creening by orcinol / H2SO4 assay show that the peptide fraction with MW 3–10 kDa has a higher carbohydrate content than the fraction with MW 5–10 kDa, and the fraction with MW below 3 kDa is less glycosylated. Therefore, the fraction with MW 3-5 kDa, dominated by [M + H]+ at m/z 3912.91, 4041.15, 4805.25 and 4936.51 is the richest in glycopeptides. The determination of glycopeptides by different mass spectrometric methods and techniques in mucus fraction with MW below 10 kDa will be the subject of a new article

Intact peptides presented in Table 1 show the presence of various amino acid residues, but mostly glycine (G), leucine (L), valine (V), proline (P), tryptophan (W), glutamic acid (E), aspartic acid (D), phenylalanine (F), and arginine (R), which are typical for peptides with established antimicrobial activities. Analyses using the ExPASy ProtParam tool indicate that peptides in Table 1 (fractions with MW <3kDa) contain both cationic and anionic and neutral peptides. Most of the peptides identified in fractions with MW <3kDa are characterized by an amphipathic structure and display generally hydrophobic surfaces, only two peptides - hydrophilic (â„–s 2 and 11). The peptides (shown in Table 1) as well as, detected previously peptides contain high levels of glycine and leucine residues belong to a new class of Gly/Leu-rich antimicrobial peptides, which passes antimicrobial activity against Gram- bacteria [Sousa et al., 2009]. 

Smith A.M.; Quick T.J.; St Peter R.L. Differences in the Composition of Adhesive and Non-Adhesive Mucus from the Limpet Lottia limatulaBiol. Bull. 1999, 196, 34–44.

Trapella C.; Rizzo R.; Gallo, S. et al. HelixComplex snail mucus exhibits pro-survival, proliferative and pro-migration effects on mammalian fibroblasts. Sci Rep 2018, 8, 17665.

Matusiewicz, M.; Kosieradzka, I.; Niemiec, T.; Grodzik, M.; Antushevich, H.; Strojny, B.; GoÅ‚Ä™biewska, M. In Vitro Influence of Extracts from Snail Helix aspersa Müller on the Colon Cancer Cell Line Caco-2. Int. J. Mol. Sci. 2018, 19, 1064.

Sousa, J.C.; Berto, R.F.; Gois, E.A.; Fontenele-Cardi, N.C.; Honório-Júnior, J.E.; Konno, K.; Richardson, M.; Rocha, M.F.; Camargo, A.A.; Pimenta, D.C.; et al. Leptoglycin: A new Glycine/Leucine-rich antimicrobial peptide isolated from the skin secretion of the South American frog Leptodactylus pentadactylus (Leptodactylidae). Toxicon 200954, 23–32.

Reviewer 2 Report

The submitted article by Topalova et al. deals with the antibacterial properties of a protein isolated from slime mucus. The article is interesting and broadly describes various studies. However, I would like to point out some issues that could improve the manuscript. 
1) The discussion of the results could include more specific references to results previously obtained by other researchers. There is no indication whether the isolated substance has higher, lower or comparable activity to proteins already studied. It would also be useful to refer to the antimicrobial activity of peptide antibiotics, e.g. polymyxins, to be able to estimate whether the isolated substance has similar/different properties. 
2) The figure showing the AFM results should also include pictures of control samples. 
3) It might be useful to relate the predicted antibacterial properties presented in Table 1. to the results obtained experimentally.

Author Response

DearEditor,

Thank you very much for your deep interest to our paper and for all remarks and comments from Reviewers. We made the necessary improvements and are sending our corrected paper and explanation of your questions.  

Reviewer 2:  Comments and Suggestions for Authors

The submitted article by Topalova et al. deals with the antibacterial properties of a protein isolated from slime mucus. The article is interesting and broadly describes various studies. However, I would like to point out some issues that could improve the manuscript. 

Reviewer: 1) The discussion of the results could include more specific references to results previously obtained by other researchers. There is no indication whether the isolated substance has higher, lower or comparable activity to proteins already studied. It would also be useful to refer to the antimicrobial activity of peptide antibiotics, e.g. polymyxins, to be able to estimate whether the isolated substance has similar/different properties. 

Answer: Thank you for the recommendation. we have not conducted such studies. The text has been rewritten and other studies previously received from other researchers have been included in the discussion of the results. Comparative studies of antimicrobial activity against Escherichia coli NBIMCC 8785 between polymyxins and peptide fraction witt MW <10 kDa would be interesting, but Ñ‚he research in this article focuses on the mechanism of antibacterial activity of some peptides. We are working on a comparative analysis of all peptides tested so far, which have been synthesized to prove the synergistic effect.

Reviewer: 2) The figure showing the AFM results should also include pictures of control samples. 

Answer: We revised the manuscript. We decided to remove the AFM analyses and to include them in further studies that will show in detail the changes of the bacterial envelops during its exposure to the antibacterial peptides. We think that this will make the manuscript more concise and the data from the different parts of it more unified.

Reviewer:  3) It might be useful to relate the predicted antibacterial properties presented in Table 1. to the results obtained experimentally.

Answer: Probably, the observed antibacterial activity against E.coli NBIMCC 8785 can be attributed peptides presented in Table 1 (â„– 1, 11-13, 15 and 16) for which was predicted high antimicrobial activity by iAMPpred-software and was detected high homology with known AMPs with activity against Gram- bacteria, such as Leptoglycin, Microcin, Ctenidin and Acanthoscurrin (See SI 1). However, based on previous research [Ilieva et al., 2020], our hypothesis is that the antibacterial activity of mucus fraction with Mw<10 kDa is due to a synergistic effect of peptides. Trapella et al., 2018, also suggest that snail mucus potential (HelixComplex) as therapeutic agent at wound repair are attributable a synergist activity of several molecules [Trapella et al., 2018].

Round 2

Reviewer 1 Report

The authors have addressed all the points successfully.